# The Relationship between Visceral Adiposity and Nonalcoholic Fatty Liver Disease Diagnosed by Controlled Attenuation Parameter in People with HIV: A Pilot Study

**DOI:** 10.3390/diagnostics12112590

**Published:** 2022-10-26

**Authors:** Giada Sebastiani, Nathalie Paisible, Cecilia Costiniuk, Joseph Cox, Dana Kablawi, Marina B. Klein, Nadine Kronfli, Jean-Pierre Routy, Julian Falutz, Bertrand Lebouché, Giovanni Guaraldi

**Affiliations:** 1Division of Gastroenterology and Hepatology, McGill University Health Centre, Montreal, QC H4A3J1, Canada; 2Chronic Viral Illness Service, McGill University Health Centre, Montreal, QC H4A3J1, Canada; 3Department of Family Medicine, McGill University, Montreal, QC H4A3J1, Canada; 4Department of Medical and Surgical Sciences for Children and Adults, University of Modena and Reggio Emilia, 41100 Modena, Italy; 5Modena HIV Metabolic Clinic, Azienda Ospedaliero-Universitaria di Modena, 41100 Modena, Italy

**Keywords:** visceral adipose tissue, transient elastography, dual-energy X-ray absorptiometry, correlation, area under the curve

## Abstract

Background: Fat alterations are frequent in people with HIV (PWH) and predict worse cardiometabolic outcomes. Visceral adipose tissue (VAT) is associated with ectopic fat accumulation in the liver. We aimed to investigate nonalcoholic fatty liver disease (NAFLD) diagnosed by controlled attenuation parameter (CAP) as a potential marker of visceral adiposity in PWH. Methods: We conducted a prospective pilot study of HIV mono-infected patients undergoing metabolic characterization and paired CAP measured by transient elastography with dual-energy X-ray absorptiometry (DEXA) scan. NAFLD was defined as CAP ≥ 285 dB/m, in absence of alcohol abuse. Excess visceral adiposity was defined as VAT > 1.32 Kg. Pairwise correlation, area under the curve (AUC) and logistic regression analysis were employed to study the association between VAT and CAP. Results: Thirty patients were included, of whom 50% had NAFLD. CAP was correlated with VAT (r = 0.650, *p* < 0.001) measured by DEXA scan. After adjusting for duration of HIV infection, body mass index and waist circumference, CAP remained the only independent predictor of excess VAT (adjusted odds ratio 1.05, 95% confidence interval [CI] 1.01–1.10). The AUC analysis determined CAP had excellent performance to diagnose excess VAT (AUC 0.92, 95% CI 0.81–1.00), higher than BMI and waist circumference. The optimized CAP cut-off to diagnose excess VAT was 266 dB/m, with a sensitivity of 88.3% and a specificity of 84.6%. Conclusions: NAFLD diagnosed by CAP is associated with VAT in PWH independently of anthropometric measures of obesity. CAP may be a potential diagnostic marker of visceral adiposity in the practice of HIV medicine.

## 1. Introduction

Since the introduction of effective antiretroviral therapy (ART), people infected with the human immunodeficiency virus (HIV) live longer [1]. However, aging people with HIV (PWH) face high rates of metabolic dysfunction due to high frequency of abdominal obesity, diabetes, dyslipidemia, hypertension and HIV-specific contributors (chronic inflammation and immune activation, gut microbiome disturbances, and ART toxicities) [2,3]. Obesity and visceral adiposity are now common in PWH. As an integral component of these frequent metabolic alterations, nonalcoholic fatty liver disease (NAFLD) has emerged as the most frequent chronic liver disease in PWH in absence of viral hepatitis coinfection and alcohol abuse, with a prevalence of 30–40% [4,5]. NAFLD is defined as a fat infiltration >5% of the liver weight in absence of other causes of hepatic steatosis [6]. Recently, a new definition of metabolic-dysfunction-associated fatty liver disease was proposed, not requiring the exclusion of secondary causes of liver diseases [7]. Thus, NAFLD may coexist with HIV [8,9].

Visceral adipose tissue (VAT) is an important compartment of body fat tissue releasing bioactive molecules and hormones such as adiponectin and leptin [10]. As a hormonally active tissue, VAT crucially contributes to the pathogenesis of obesity-related disorders. VAT is associated with cardiovascular risk, insulin resistance, and metabolic unhealthy obesity, and with ectopic fat accumulation in the liver, and as such visceral adiposity and NAFLD are pathogenetically linked [11,12]. The distribution of lipids between VAT and liver tissue has been implicated in metabolic hemostasis and progression of NAFLD. The DEXA scan is considered the gold standard for measuring visceral fat [12]. As visceral obesity is associated with poor prognosis, metabolic disturbances and degree of pathology in several chronic diseases, it is of great importance to identify methods that quantify adipose tissue accurately and can serve as markers of VAT. Controlled attenuation parameter (CAP) is a novel method for the non-invasive assessment of hepatic steatosis, which measures the increased attenuation of ultrasound waves when travelling through steatotic hepatic tissue, compared to normal liver tissue. CAP has demonstrated an excellent diagnostic accuracy to quantify hepatic steatosis against the gold standard liver biopsy, and also magnetic-resonance-imaging-derived proton density fat fractions in PWH, with an area under the receiver operating characteristic curve (AUC) of 0.88 and 0.82, respectively [13,14]. CAP is conveniently performed concomitantly with liver stiffness measurement during transient elastography, a non-invasive tool validated in the specific setting of NAFLD in PWH [15,16].

In this prospective pilot trial, we aimed to: (1) investigate the association between NAFLD diagnosed by CAP and visceral fat diagnosed by a DEXA scan; and (2) determine the patients’ acceptance of a one-day comprehensive metabolic evaluation.

## 2. Materials and Methods

### 2.1. Study Design

We conducted a prospective observational trial of consecutive PWH enrolled at one single center, the McGill University Health Centre (MUHC) in Montreal, Canada. Patients underwent a one-day comprehensive metabolic evaluation, including CAP measured by transient elastography, DEXA scan, anthropometric evaluation, blood test and standardized study questionnaires from the LIVEr disease in HIV Cohort [17,18]. We included all patients with HIV infection (as documented by positive enzyme-linked immunosorbent assay [ELISA] with Western blot confirmation) aged ≥18 years and on stable ART for at least 6 months with suppressed HIV infection (HIV viral load <40 copies/mL). Exclusion criteria were: (i) positivity for hepatitis C virus antibody or hepatitis B surface antigen; (ii) evidence of other liver disease; (iii) significant alcohol intake, defined by an Alcohol Use Disorders Identification Test (AUDIT-C) questionnaire score of ≥4 for men and ≥3 for women [19]; (iv) history of hepatocellular carcinoma or liver transplantation; (v) contraindications (pregnancy, pacemaker insertion) and failure or unreliable transient elastography examination (less than 10 valid measures or interquartile range >30%) [20].

The study was approved by the Research Ethics Board of the Research Institute of MUHC (code 2021-6656) and was registered at ClinicalTrials.gov (NCT 05359471). The study was conducted according to the Declaration of Helsinki and Good Clinical Practice guidelines. All patients provided their informed written consent prior to participation.

### 2.2. Clinical and Biological Parameters

The primary study outcome was the presence of excess visceral fat, defined as VAT ≥ 1.32 Kg by DEXA scan examination. The software algorithm worked through detection of the width of subcutaneous fat layer within the android region of interest on the lateral part of the abdomen and the interior–posterior thickness of the abdomen, which can be assessed using X-ray attenuation. The results of fat-free mass, VAT/total body fat and VAT/weight were expressed in kilograms and percentage. All DEXA measurements were carried out at the Centre for Innovative Medicine of MUHC.

We used questionnaires to collect demographic information, HIV and medication history. A physical examination with measurement of body mass index (BMI) and waist circumference was conducted. A same-day blood test was performed for liver biochemistries, lipid profile, hematological and immuno-virological parameters. Type 2 diabetes mellitus was defined as a glycosylated hemoglobin of 6.5% or greater, or as previously diagnosed by an endocrinologist/treating physician. Lipid accumulation product was calculated as follows: (waist circumference–65) × triglycerides [mmol/L] in males; and (waist circumference–58) × triglycerides [mmol/L] in females. Lipid accumulation product has been proposed as biomarker of central fat accumulation and indicator of the risk of insulin resistance, metabolic syndrome, type 2 diabetes, and cardiovascular disease [21].

Examinations were performed on a 4 h fasting patient by maximum two experienced operators at each site (>500 examinations before the study). The standard M probe was used in all patients. The XL probe was used in case of failure of M probe and if BMI > 30 Kg/m^2^. The following criteria were applied to define the result of liver stiffness measurement as reliable: at least 10 validated measures and an interquartile range <30% of the median [20]. A CAP cut-off of 285 dB/m was used to define NAFLD [14].

### 2.3. Statistical Analysis

The sample size calculation was performed for a pilot study measuring a proportion. There is little published guidance on sample size for pilot trials [22] and the justification can be based on a number of methods [23]. Browne gives a general rule to take a minimum of 30 patients to estimate a parameter [24]. Thus, the estimated sample size for this trial was 30 PWH.

We compared characteristics of participants by outcome status using Student’s *t*-test for continuous variables and Pearson’s χ2 for categorical variables. Correlation coefficients of VAT with CAP, BMI, waist circumference and lipid product accumulation were calculated using the Pearson correlation analysis. Predictors of excess VAT were determined using unadjusted and adjusted logistic regression models and reported as adjusted odds ratios (aOR) with 95% confidence interval (CI). The adjusted regression model included covariates determined a priori to be clinically important. The diagnostic performance of CAP, BMI and waist circumference to predict excess VAT was measured as AUC. The diagnostic performance was considered acceptable for AUCs between 0.80 and 0.90 and excellent if >0.90. The diagnostic performances were also assessed in terms of sensitivity, specificity, and the likelihood ratio for a positive test result (LR+) and the LR for a negative test result (LR-). The optimal cut-off value of CAP to diagnose excess VAT was selected to maximize the sum of sensitivity and specificity. All tests were two-tailed and with a significance level of α = 0.05. Statistical analyses were performed using STATA 15 (STATA Corp. LP, College Station, TX, USA).

## 3. Results

Between December 2021 and April 2022, 33 HIV mono-infected patients were screened for the study based on the inclusion and exclusion criteria, of whom 30 (90.9%) agreed to be included. Reasons to decline study participation included fear of coming to the hospital in the COVID-19 pandemic, planning pregnancy and living far from the MUHC. Of the 30 patients included, 27 (90%) were male, the mean age was 48.5 (standard deviation [SD] 13.2) years, and 8 (26.7%) had type 2 diabetes. Mean BMI and waist circumference were 29.9 (SD 4.8) Kg/m^2^ and 100.9 (9.2) cm, respectively. Compared to PWH without NAFLD, the 15 PWH with NAFLD (50%) had higher VAT (2.06, SD 0.82 vs. 1.10, SD 0.82 Kg; *p* = 0.003) and VAT/total body weight ratio (0.078, SD 0.032 vs. 0.041, SD 0.021%; *p* < 0.001) but no difference in fat mass (27.9, SD 9.4 vs. 25.7, SD 9.6 Kg; *p* = 0.54). Table 1 reports the characteristics of the study population and univariable analysis by excess VAT status.

When compared to those without excess VAT, PWH with excess VAT were older, had longer duration of HIV infection and had higher BMI and waist circumference. PWH with excess VAT had more frequent history of cardiovascular events, higher triglycerides, lower HDL cholesterol and higher lipid accumulation product. Finally, both CAP and liver stiffness measurement were higher in PWH with excess VAT. CAP positively correlated with all visceral fat measurements, including VAT (r = 0.650, *p* < 0.001), VAT/body weight ratio (r = 0.565, *p* = 0.001) and fat mass (r = 0.390, *p* = 0.033). Both BMI and waist circumference showed correlation with VAT and fat mass, but not with VAT/body weight ratio (Figure 1).

After adjusting for duration of HIV infection (aOR 1.01 per year, 95% CI 0.91–1.12; *p* = 0.921), BMI (aOR 1.77, 95% CI 0.74–4.23; *p* = 0.202) and waist circumference (aOR 0.91 per cm, 95% 0.68–1.21; *p* = 0.509), CAP remained the only independent predictor of excess VAT (aOR 1.05 per dB/m, 95% CI 1.01–1.10; *p* = 0.036).

The AUC analysis determined CAP had excellent performance to diagnose excess VAT (AUC 0.92, 95% CI 0.81–1.00). BMI (AUC 0.83, 95% CI 0.68–0.99) and waist circumference (AUC 0.81, 95% CI 0.65–0.97) also had good diagnostic performance (Figure 2).

Due to the relative small sample size, we did not run a direct comparison among CAP, BMI and waist circumference. The optimized cut-off of CAP to diagnose excess VAT was 266 dB/m, able to correctly classify 86.7% of patients, with a sensitivity of 88.3% and a specificity of 84.6%, LR + 5.74, LR − 0.14.

## 4. Discussion

In this pilot trial in PWH, we found that CAP, a non-invasive test to diagnose NAFLD, is associated with measures of visceral adiposity diagnosed by the DEXA scan, the gold standard for measuring visceral fat. This association was independent of anthropometric measures of obesity such as BMI and waist circumference. CAP also showed an excellent AUC to diagnose excess VAT, with an optimal cut-off identified at 266 dB/m. Since VAT accumulation is associated with ectopic fat accumulation in the liver, the presence of NAFLD may be considered as a mirror for visceral fat, carrying an increased risk for cardiovascular and overall mortality. This pilot study also showed the feasibility of conducting a one-day point-of-care comprehensive metabolic program, as 90.9% of PWH consented to be part of the trial. Our pilot findings suggest that CAP, a convenient test performed simultaneously with liver stiffness measurement during transient elastography, may be a marker of visceral fat beyond the liver.

Visceral fat is a key element in the development of multiple non-AIDS comorbid disease states, including cardiovascular disease and liver disease, which became leading causes of death among PWH in the ART era [11]. The development of age-related comorbidities is driven by metabolic-dysfunction and HIV-specific contributors [25]. Obesity, and particularly VAT accumulation, are associated with systemic and adipose tissue inflammation, dyslipidemia, insulin resistance and excess oxidative stress. Aging is associated with paraphysiological central fat redistribution, chronic inflammation and adipocyte senescence, all of which may be magnified in PWH. Adipose tissue is also a potential HIV reservoir [11]. As visceral fat is associated with poor prognosis, metabolic disturbances and degree of pathology in several aging-related comorbidities, it is clinically important to identify diagnostic methods that quantify adipose tissue accurately and can serve as markers of VAT.

NAFLD is emerging as the leading cause of chronic liver disease in PWH, with higher rates than the general population [4,5,16,26]. A meta-analysis of five studies reported an overall prevalence of NAFLD at 35.3% [4]. Besides commonly presenting with features of metabolic syndrome, HIV-infected patients have unique risk factors for NAFLD, including HIV-related inflammation, ART, persistent immunoactivation, and lipodystrophy [2,27]. Unhealthy metabolic status and visceral fat may have a greater impact on NAFLD than obesity itself in PWH. Indeed, about 35% of PWH with NAFLD have a BMI below 25 Kg/m^2^, denoted as lean NAFLD [18]. HIV mono-infected patients with NAFLD tend to have lower BMI compared to uninfected NAFLD patients [28]. Moreover, changes in anthropometric characteristics in PWH, such as a shift of body fat deposits from the subcutaneous to the visceral compartment, have been observed [2]. As such, visceral fat may be a key element in the pathogenesis of NAFLD in PWH independently of insulin resistance and classical anthropometric measures of abdominal obesity. Given the prognostic relevance of visceral fat beyond liver diseases, including predicting cardiovascular disease and overall mortality, identification of indirect markers of VAT may be useful in the practice of HIV medicine. CAP is a non-invasive diagnostic tool which showed good performance against liver histology for quantification of liver fat [29,30]. Ad hoc studies in the context of PWH confirmed its diagnostic value for NAFLD [14,31]. An important advantage of this surrogate for steatosis is that its measurement is contemporary with liver stiffness measurement during a transient elastography examination, providing relevant clinical information for both liver fibrosis and hepatic steatosis during a point-of-care examination. This is particularly relevant in PWH, a population at high risk for NAFLD in which guidelines recommend case-finding of liver fibrosis [32]. Our pilot study suggests that CAP may also provide information on visceral fat as a marker of VAT measured by a DEXA scan. Previous studies reported the association between CAP and VAT in HIV-uninfected NAFLD [33]. In our study, CAP correlated with VAT, and this association also holds true in the multivariable model independently of classical anthropometric measures of obesity. These pilot data suggest that CAP may have a better diagnostic value for visceral fat compared to BMI and waist circumference. This was also hinted by the AUC analysis, reporting an excellent performance for CAP rather than for BMI and waist circumference. The AUC also identified an optimized cut-off of 266 dB/m of CAP to diagnose excess VAT, with a sensitivity of 88.3% and a specificity of 84.6%. CAP also showed a strong correlation with the lipid accumulation product, a known predictor for central lipid accumulation linked to diabetes risk and cardiovascular disease [21]. Interestingly, CAP showed a weaker association with fat mass, suggesting that the link between NAFLD and visceral fat may be more complex in PWH given higher rates of lean NAFLD and sarcopenia.

Our pilot trial has several strengths, including the prospective design allowing detailed metabolic characterization and the use of an easily accessible and validated non-invasive tool to diagnose NAFLD. The main limitation of our study lies in the small sample size, which could not provide enough power for direct comparison among CAP, BMI, and waist circumference as surrogates for visceral fat. Moreover, our regression model may suffer from the same limitation. We also acknowledge that the lack of longitudinal data impedes us to speculate on any causal association of our findings. Finally, we did not have the availability of a control group in this pilot study. Our results may not apply to the HIV-uninfected population since PWH may have abnormal distribution of fat, such as HIV-associated lipodystrophy, characterized by abnormal accumulation of trunk fat, that differs clinically from that of the general population in the presence of normal BMI and sarcopenia [12]. Our study may provide rationale for a larger study, also including an HIV-uninfected control group.

## 5. Conclusions

In conclusion, our results suggest CAP as a potential marker for visceral fat in PWH. Given that guidelines recommend case-finding for NAFLD-related liver fibrosis in PWH via non-invasive tools such as transient elastography, CAP may provide synchronous information on the metabolic health status, visceral fat and cardiovascular risk. Larger studies with a HIV-uninfected control group, longitudinal design and long follow-up data are needed.

## Figures and Tables

**Figure 1 diagnostics-12-02590-f001:**
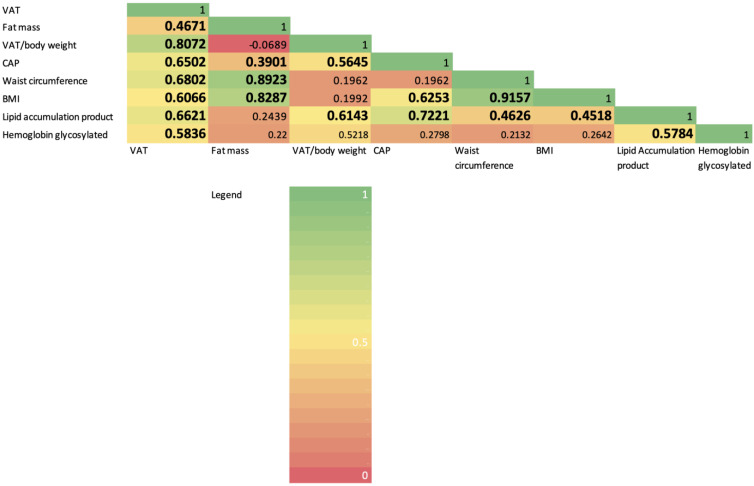
Correlation matrix with heatmap based on correlation coefficient analysis. Significant values (*p* < 0.05) are bolded.

**Figure 2 diagnostics-12-02590-f002:**
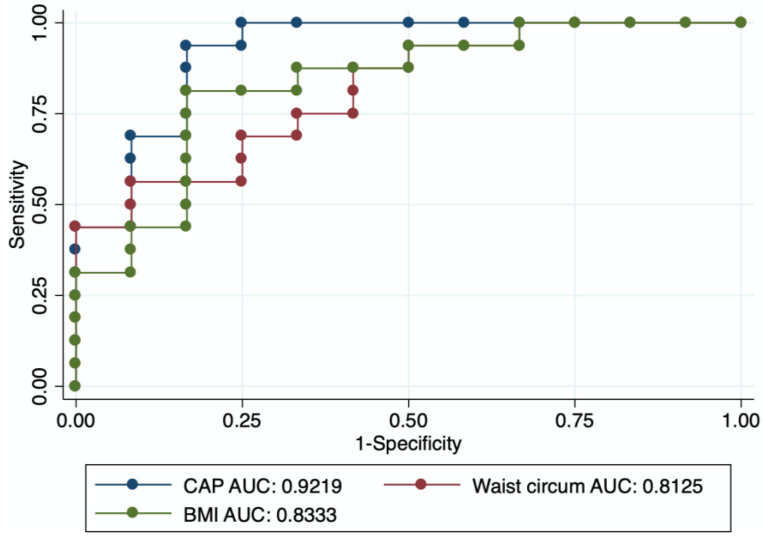
Area under the curve of CAP, BMI and waist circumference for prediction of excess VAT.

**Table 1 diagnostics-12-02590-t001:** Characteristics of the study population by excess VAT status (*n* = 30).

	Excess VAT (*n* = 17)	No Excess VAT (*n* = 13)	*p*-Value
**Age** (years)	52.9 (12.4)	42.8 (12.5)	0.035
**Male sex** (%)	16 (94.2)	11 (84.6)	0.390
**Ethnicity (%)**			
White/Caucasian	7 (41.2)	3 (23.1)	0.503
Black non-Hispanic	3 (16.7)	4 (30.8)	
Hispanic	4 (23.6)	3 (23.1)	
**Hypertension** (%)	7 (41.2)	4 (30.8)	0.558
**Diabetes** (%)	6 (35.3)	2 (15.4)	0.222
**History of cardiovascular event** (%)	5 (29.4)	0	0.032
**Waist circumference** (cm)	106.6 (10.8)	93.3 (11.8)	0.004
**BMI** (Kg/m^2^)	32.1 (4.3)	26.9 (3.8)	0.002
**Time since HIV diagnosis** (years)	20.0 (13.4)	9.4 (9.0)	0.021
**CD4 cell count** (cells/μL)	679.4 (396.6)	595.5 (315.6)	0.357
**Nadir CD4 cell count** (cells/μL)	254.2 (151.9)	214.6 (206.4)	0.684
**Current ART regimen (%)**			
NRTIs	14 (82.4)	13 (100)	0.110
NNRTIs	2 (11.8)	0	0.201
PIs	2 (11.8)	1 (7.7)	0.713
Integrase inhibitors	17 (100)	12 (92.3)	0.245
**Past exposure to d-drugs** (%)	4 (25.0)	1 (8.3)	0.254
**ALT** (IU/L)	51.9 (43.9)	26.2 (21.4)	0.118
**AST** (IU/L)	34.4 (26.4)	24.3 (7.3)	0.210
**Platelets** (10^9^/L)	242.4 (178.4)	195.5 (46.9)	0.366
**Triglycerides** (mmol/L)	2.92 (2.04)	1.04 (0.56)	0.047
**Total cholesterol** (mmol/L)	4.51 (1.12)	4.14 (0.95)	0.490
**HDL cholesterol** (mmol/L)	1.03 (0.26)	1.37 (0.28)	0.025
**LDL cholesterol** (mmol/L)	2.48 (1.11)	2.17 (0.83)	0.573
**Lipid accumulation product**	111.5 (53.3)	38.1 (36.8)	<0.001
**Fasting glucose** (mmol/L)	6.14 (1.75)	5.33 (1.24)	0.253
**Hemoglobin glycosylated** (%)	6.23 (1.32)	5.91 (1.14)	0.394
**LSM** (kPa)	7.4 (2.5)	4.8 (1.5)	0.002
**CAP** (dB/m)	319.2 (51.6)	213.1 (52.4)	<0.001

Notes: Continuous variables are expressed as median (interquartile range) and categorical variables as number (%). The *p*-values refer to Student’s *t*-test or χ^2^ test between excess VAT and no excess VAT.

## Data Availability

According to stipulations of the patient consent form signed by all study participants, the ethical restrictions imposed by our Institutional Ethics review boards (Institutional Ethics Review Board Biomedical B Research Ethics Board of the McGill University Health Centre), and the legal restrictions imposed by Canadian law regarding clinical trials, anonymized data are available upon request. Please send data access requests to Sheldon Levy, Biomedical B (BMB) Research Ethics Board (REB) Coordinator Centre for Applied Ethics, 5100, boul. de Maisonneuve Ouest, 5th floor, Office 576, Montréal, Québec, H4A 3T2, Canada.

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
