# Peer review of "The Relationship between Visceral Adiposity and Nonalcoholic Fatty Liver Disease Diagnosed by Controlled Attenuation Parameter in People with HIV: A Pilot Study"

_diagnostics, 2022, doi:10.3390/diagnostics12112590_

Round 1

Reviewer 1 Report (New Reviewer)

The authors have made an interesting study on HIV patients and investigated nonalcoholic fatty liver disease and suggested controlled attenuation parameter (CAP) as a potential marker of visceral adiposity among HIV patients.

Although the study is a pilot study with a small group of population, some of the recent studies follow the similar approach. (10.1177/2040622322110275, 10.1097/QAD.0000000000001241)

A more novelty in the approach among the study population would be of interest to the readers.

Manuscript is written clear but needs few changes before it can be accepted.

1. A more descriptive introduction would be useful for the readers to understand the concept. 

2. In the methods section, description with subtitles such as Study population, ..., Statistical analysis, etc. would make the concept more clear.

3. In the discussion, novelty of the approach in terms of Canadian population if any can be described.

Author Response

Reviewer 2 Report (Previous Reviewer 1)

I agree with the revision except one aspect: The Term "HIV-associated NAFLD" or "NAFLD in people with HIV" should be commented somewhere. As mentioned earlier, for many hepatologists the NAFLD diagnosis requires exclusion of secondary causes of steatosis (while HIV is a known steatogenic virus). So NAFLD is seen as a diagnosis of exclusion. As this term seems to be used in the literature, a small comment would help to prevent confusion by readers.

Author Response

This manuscript is a resubmission of an earlier submission. The following is a list of the peer review reports and author responses from that submission.

Round 1

Reviewer 1 Report

The authors submitted a manuscript of a small study sample investigating the association of controlled attenuation parameter (CAP) measured by transient elastography and visceral adiposity measured by DEXA scan in people with HIV (PWH). While I congratulate the authors on conducting such a study and proving the feasibility I am not convinced of the novelty and impact of this investigation. Some important aspects (e.g., methodology, deep literature overview) are missing. However, the manuscript is quite well written and the bottom line message of the manuscript might in fact be useful for clinicians if it holds true.  I made a few suggestions and I am happy to review the manuscript again should the authors decide to resubmit it.

Aspects:

-         There are several conflicts in terms of terminology: (i) The term “HIV-associated nonalcoholic fatty liver disease (NAFLD)” is new to me as the current NAFLD definition requires the exclusion of other causes of steatosis – however, HIV is a known steatogenic virus and established secondary cause of steatosis. (ii) Transient elastography (= FibroScan) is the method with with liver stiffness and CAP can be measured. This is mixed up at several locations in the manuscript. Iii) the concept of “metabolic dysfunction” (and the newly suggested definition of MAFLD instead of NAFLD) is not mentioned at all. Instead of writing about “conditions of metabolic dysfunctions” the authors use unusual terminology (“traditional conditions”) to summarize obesity, diabetes etc.

-         Methodology not clearly described: i) which questionnaires were used? Were they standardized? Ii) Which exclusion criteria for a reliable transient elastography measurement were used? Iii) which “complete anthropometry evaluation” was performed? I only read information regarding BMI and waist circumference.

-         Statistical evaluation: Given the small sample size it is unclear whether a t test can be used (i.e., are the presented results still significantly different if other measures such as a Mann-Whitney-U test are used?). Using regressing models with several covariates is prone for overfitting. Providing medians with IQR appears more appropriate than mean with SD.

-         There cited literature is not of much help. For instance, no study citing the validity of transient elastography in PWH is cited. This proof of concept is necessary before initiating the study concept at hand. Moreover, the rationale for the cut-offs for CAP and VAT are not clearly demonstrated (e.g., a CAP cut-off of 285 dB/m is rather unusual). Moreover, the authors write about guidelines recommending the use of TE in PWH without providing the according reference.

-         I would recommend to “tune down” the phrasing at several sites. For example, describing the “detailed metabolic characterization” as a strength is not supported by the information provided in the manuscript.

Reviewer 2 Report

This is an interesting topic. However, the small sample size as also discussed by the Authors does not provide enough power for direct comparison among CAP, BMI, waist circumference as surrogates for visceral fat.  Therefore, CAP is not a clear diagnostic marker of visceral adiposity as instead, stated in the Title. Only a more extensive study on such an important issue could add sufficient information. Additionally, the study should be performed firstly in HIV negative patients, or at least such a control group should be added, since HIV carriers may have abnormal distribution of fat (HIV-associated lipodystrophy characterized by abnormal accumulation of trunk fat, including VAT) that differs clinically from that of general population (i.e., in presence of normal BMI, sarcopenia, etc.) 

Reviewer 3 Report

Giada Sebastiani et al. present an easily accessible and validated non-invasive marker CAP of visceral adiposity to diagnose HIV-associated nonalcoholic fatty liver disease (NAFLD).The manuscript was based on a pilot study,which had a neoteric perspective,and showed a strong association with CAP and visceral adiposity diagnosed by DEXA scan ,which was independent beyond BMI and waist circumference. That was instructive.However, there was lack of adequate,evidential clinical sample size to directly support all the conclusions of the manuscript,especially the advantage of CAP over BMI or waist circumference. Moreover,regarding the studies in recent years,more current and relevant references should be cited. Besides,throughout the manuscript,eventhough the presentation is proper and easy to understand, there are still some terms,expressions which need to be polished and several English spelling or grammar errors that should be refined.